# Evaluation of quantitative biosensor for glucose-6-phosphate dehydrogenase activity detection

**Pairat Pengboon[1], Areenuch Thamwarokun[2]ᵒ, Khaimuk Changsri[3]ᵒ, Chollanot Kaset[3], Sirinart Chomean[3]***

**1** Graduate Program in Medical Technology, Faculty of Allied Health Sciences, Thammasat University, Pathumtani, Thailand, **2** Graduate Program in Biomedical Sciences, Faculty of Allied Health Sciences, Thammasat University, Pathumtani, Thailand, **3** Department of Medical Technology, Faculty of Allied Health Sciences, Thammasat University, Pathumtani, Thailand

ᵒ These authors contributed equally to this work.
* sirinat.c@allied.tu.ac.th

**Data Availability Statement:** All relevant data are within the manuscript and its Supporting Information files.

## Abstract

Neonatal jaundice is a common and severe disease in premature infants with Glucose-6-Phosphate Dehydrogenase (G-6-PD) deficiency. The World Health Organization (WHO) has recommended screening for G-6-PD deficiency in newborns for early recognition as well as to prevent unwanted outcomes in a timely manner. The present study aimed to assess a point-of-care, careSTART™ G6PD biosensor as a quantitative method for the diagnosis of G-6-PD deficiency. Factors influencing the evaluation of G-6-PD enzyme activity were examined in 40 adults, including ethylenediaminetetraacetic acid (EDTA) anticoagulant, hematocrit concentration, storage temperature and time. Analytic performance of the careSTART™ G6PD biosensor was evaluated in 216 newborns and compared with fluorescent spot test (FST) and standard quantitative G-6-PD enzyme activity (SGT) assay. The results of factors affecting the G-6-PD enzyme activity showed that the activity determined from finger-prick was not statistically different from venous blood ($p = 0.152$). The G-6-PD value was highly dependent on the hematocrit and rose with increasing hematocrit concentration. Its activity was stable at 4°C for 3 days. Reliability analysis between the careSTART™ G6PD biosensor and SGT assay showed a strong correlation with a Pearson's correlation coefficient of 0.82 and perfect agreement by intraclass correlation coefficient (ICC) of 0.90. Analysis of the area under the Receiver Operating Curve (AUC) illustrated that the careSTART™ G6PD biosensor had 100% sensitivity, 96% specificity, 73% positive predictive value (PPV), 100% negative predictive value (NPV) and 97% accuracy at 30% of residual activity. While the diagnostic ability for identifying G-6-PD deficiency had 78% sensitivity, 89% specificity, 56% positive predictive value (PPV), 96% negative predictive value (NPV) and 88% accuracy when stratified by gender. The careSTART™ G6PD biosensor is an attractive option as a point-of-care quantitative method for G-6-PD activity detection. Quantification of G-6-PD enzyme activity in newborns is the most effective approach for the management of G-6-PD deficiency to prevent severe jaundice and acute hemolysis.

**Funding:** This study was supported by Thammasat University Research Fund (Contract No. TUGG138/2562) which cover the chemical and reagents supply. The careSTARTTM G6PD biosensor was provided by the manufacturer (WELLS BIO, INC., Korea). However, the funders had no role in study design, data collection and analysis, decision to publish, or preparation of the manuscript.

**Competing interests:** The authors have declared that no competing interests exist.

## Introduction

Glucose-6-phosphate dehydrogenase (G-6-PD) deficiency affects millions of people worldwide and comprises the most common inherited blood disorder [1–3]. Defective G-6-PD enzymes cause the increased susceptibility of red blood cells to reactive oxygen species, leading to hemolysis [1, 4]. Its clinical manifestations vary from asymptomatic to severe hemolytic anemia depending on the residual G-6-PD enzyme activity [1, 5]. Affected individuals generally present with acute hemolytic anemia, favism, or chronic non-spherocytic hemolytic anemia [1]. The clinical symptoms tend to increase as the red blood cells undergo oxidative stress triggered by agents such as anti-malarial drugs (primaquine), mothballs, infection or the ingestion of fava beans [1, 3–9]. Several studies have reported that diabetes, myocardial infarction and strenuous physical exercise can stimulate hemolysis in individuals with G-6-PD deficiency [1, 7]. Notably, affected neonates frequently show severe jaundice. Lack of awareness concerning G-6-PD deficiency in newborns could cause extreme hyperbilirubinemia, bilirubin neurotoxicity, kernicterus and, eventually, mental retardation [8, 10]. Accordingly, the World Health Organization (WHO) has recommended screening for G-6-PD deficiency in newborns to promote early diagnosis and prevent unwanted outcomes in a timely manner [8, 10–13]. Moreover, individuals suffering from severe G-6-PD deficiency with residual activity less than 30% and 70% should be excluded from primaquine and tafenoquine administration, respectively [14, 15]. Thus, the measurement of G-6-PD activity is a crucial step before starting malaria treatment. Although the fluorescent spot test (FST) is widely used for qualitative screening of G-6-PD deficiency, its discriminative power appears limited, requiring UV visualization and highly specific skills for interpretation [6, 8, 16–19]. The reference method for G-6-PD deficiency detection is based on a quantitative G-6-PD enzymatic assay. However, it is time-consuming, laborious and requires a spectrophotometer, which may not be suitable for the field or large-scale testing [17, 18]. Quantitative point-of-care G-6-PD tests are an essential tool in low-resource settings. Recently, the point-of-care quantitative careSTART[TM] G6PD biosensor was developed as an alternative to the labor-intensive standard G-6-PD enzymatic method.

The careSTART[TM] G6PD biosensor is an electrochemical biosensor that measures the electron transfer from the change of nicotinamide adenine dinucleotide phosphate (NADPH) into its reduced form by using G-6-PD enzymes. It is proposed for the quantitative measurement of G-6-PD enzyme activity in whole blood. Therefore, the present study aimed to examine the performance of the careSTART[TM] G6PD biosensor compared with the gold standard enzymatic assay. Moreover, the most widely used FST method was evaluated. Measurement interference effects were also investigated, including ethylenediaminetetraacetic acid (EDTA), hematocrit concentration and enzymatic stability.

## Materials and methods

### Sample collection

Ethical approval for this study was obtained from the Third Ethics Committee of Thammasat University, Pathum Thani Province, Thailand (COA No. 150/2561). Drawing blood from a newborn can be difficult, potentially harmful and may yield limited sampling blood volume. Therefore, the factors affecting the G-6-PD enzyme activity measurement were evaluated in 40 adults. Prior to enrolment, all volunteers provided informed written consent to participation in the study as well as publication of the results. The volunteers included 20 males and 20 females ranging in age between 18 and 24 years old. The blood samples were collected from both finger-prick (5 µL) and venous puncture (3 mL in $K_2$EDTA vacutainer tubes), after which the samples were analyzed by standard quantitative G-6-PD enzymatic assay (OSMMR2000-D G-6-PD kit, R&D Diagnostics, Ltd., Greece).

To evaluate the performance of the careSTART[TM] G6PD biosensor (WELLS BIO, INC., Korea), 216 neonatal blood samples ranging in age between 1 to 35 years old were collected and kept in a microtainer tube with $K_2EDTA$ anticoagulant. The ethics committee provided a waiver of informed consent due to the neonatal blood samples used in this study being leftover samples following routine analysis. All data samples were de-identified before access. Complete blood count (CBC) (DxH 800 hematology analyzer, Beckman Coulter, Inc., USA) and fluorescent spot test (FST) (SQMMR500, R&D Diagnostics, Ltd., Greece) were assessed to screen for G-6-PD deficiency. G-6-PD enzyme activity was measured by the careSTART[TM] G6PD biosensor compared with standard quantitative G-6-PD enzymatic assay (SGT).

All blood samples were carried out a single time following standard operating procedures under controlled time and temperature.

## Assessment of factors influencing G-6-PD enzyme activity

Capillary blood samples were immediately assessed for G-6-PD enzyme activity by the standard SGT. Meanwhile, venous blood was aliquoted into 3 parts for assessment of the factors that affect G-6-PD enzyme activity, including anticoagulant (EDTA), hematocrit (Hct) concentration and enzymatic stability.

The effect of EDTA anticoagulant collected from venous blood was investigated by measuring G-6-PD enzyme activity and comparing with the sample from finger-prick.

Due to the activity of G-6-PD enzyme depending on the storage time and temperature, EDTA blood was aliquoted into a sealed, dark 1.5 mL microcentrifuge tube and kept at room temperature (25°C) and 4°C for 3 days. G-6-PD enzyme activity was measured at 1, 2, 6, 12, 24, 48 and 72 hours.

The proportion of red blood cells to total blood volume (Hct) varies substantially depending on race, age and sex, which are commonly used to determine the presence of anemia. In this study, the Hct levels of each of the 40 samples were adjusted to 30%, 40%, 50% and 60% by using their own plasma. Each Hct level was assessed for G-6-PD enzyme activity and compared to each other.

The study of factors influencing G-6-PD enzyme activity was performed using the SGT assay according to the manufacturer's instructions. Briefly, five microliters of blood were mixed with the Elution Buffer (75 μL) in a U-bottom microtiter plate for 10 min at room temperature (25±2°C). The eluted (15 μL) blood was transferred to 75 μL of the Reagent Mixture contained in a new flat-bottom microtiter plate and mixed thoroughly. Then, 100 μL of Color Reagent Mixture was added to each well and G-6-PD activity reaction (kinetic mode) was read with an ELISA reader (800TS Microplate Reader, BioTek, USA) at 550 nm for 15 min at 1 min intervals. After the final reading was taken, the plate was read at wavelength 405 nm to get the hemoglobin (Hb) content of each sample. G-6-PD enzyme activity was calculated by comparing the rates of a blood sample to the rate of normal control with known G-6-PD activity. The reported G-6-PD activity was normalized to Hb and the results expressed directly into international units per gram of hemoglobin (IU/gHb).

All experiments were conducted for the quality control of deficient blood (2.0 IU/g Hb with the specification of 1.0–3.0 IU/gHb) and normal blood (14.6 IU/gHb with the specification of 9.5–19.7IU/gHb in duplicate.

## Performance of the careSTART[TM] G6PD biosensor for G-6-PD enzyme activity detection

The diagnostic ability of the quantitative careSTART[TM] G6PD biosensor was tested using 216 neonatal blood samples. According to the manufacturer's instructions, ten microliters of blood pricked by lancet were applied to the end of the test strip, which was inserted into the analyzer.

After 4 mins, the G-6-PD enzyme activity result was displayed on the analyzer's monitor in terms of units per deciliter (IU/dL). The activity derived from the biosensor measured the G-6-PD enzymes from RBC in one deciliter, which should be divided by Hb (IU/gHb). The EDTA blood was also used for identifying G-6-PD deficiency by qualitative FST assay. Five microliters of EDTA blood was added to 100 μL of G-6-PD substrate reagent, mixed thoroughly and incubated at 25˚C for 10 min. After that, the mixture (100 μL) was applied to filter paper and observed under UV light. The results acquired by the careSTART[TM] G6PD biosensor and FST were compared with standard quantitative G-6-PD assay (SGT) as a reference assay.

## Statistical analysis

Statistical analysis was performed using MedCal software (version 18.2.1; MedCalc, Mariakerke, Belgium) and IBM SPSS statistics (IBM SPSS Statistics for Windows, Version 22.0. Armonk, NY: IBM Corp software). Since all data were normally distributed, the parametric statistics test was used throughout the study. The mean, median, standard deviation (SD) and ranges were determined for all G-6-PD enzyme activity values with a 95% confidence interval (CI). Normal G-6-PD activity or 100% activity can be defined by the median value of hemizygous males with a G-6-PD normal allele, called adjusted male median (AMM), which is calculated according to the description of Domingo GJ et al. [20]. Any males and females who have G-6-PD activity less than to 30% of the AMM must be regarded G-6-PD deficient [20–22]. Meanwhile, males with G-6-PD activity of 30% or more of the AMM can be identified as G-6-PD normal. Heterozygous females can have intermediate status with G-6-PD activity levels ranging from 30% to 80% of AMM. A G-6-PD activity value of 80% or more of the AMM is considered G-6-PD normal [20–22]. Also, $p$ values less than 0.05 indicate a statistically significant difference between the groups. A paired t-test was used to evaluate the difference of G-6-PD enzyme activity measured from finger-prick and venous blood keeping in EDTA vacutainer tube. The effect of Hct concentration and storage time was assessed by one-way ANOVA running with a post hoc test. A linear regression model was used to predict the time at which G-6-PD enzymatic activity fell below 90% of the median initial value. Pearson correlation (r) and Bland-Altman plots were performed to evaluate the relationship, the degree of agreement between careSTART[TM] G6PD biosensor and the reference SGT assay. Analysis of the area under the Receiver Operating Curve (AUC) under the receiver operating characteristic (ROC) curve was performed between the careSTART[TM] G6PD biosensor and a quantitative SGT assay at 30%, 70% and 80% of AMM. The performance of the careSTART[TM] G6PD biosensor included sensitivity, specificity, positive likelihood ratio (+LR), negative likelihood ratio (-LR), positive predictive value (PPV), negative predictive value (NPV) and disease prevalence was tested applying standard formulas [23]. The diagnostic potential of the careSTART[TM] G6PD biosensor and FST analyzed in clinical samples was determined between genders, assuming the SGT as the reference method.

## Results

### Study population and distribution of G-6-PD enzyme activity

Study population characteristics including mean, median and range of G-6-PD activity were stratified by gender (Table 1). G-6-PD enzyme activity measured by quantitative SGT was taken for normal distribution. Forty adult samples from males and females with an average age of 21 (21 ± 1 year) and Hct level of 42% (42 ± 3.7%) were studied. The results demonstrated that there were no significant differences in enzyme activity between females (5.8 ± 1.9 IU/gHb) and males (6.4 ± 2.1 IU/gHb) ($p$ = 0.328). The adult adjusted male median was 6.6 IU/

**Table 1. Proposed reference values for G-6-PD enzyme activity in this study population.**

| Reference values | Total | Female | Male | Adjusted male |
|---|---|---|---|---|
| **Adult** | | | | |
| Number of cases | 40 | 20 | 20 | 20 |
| Mean (95% CI; IU/gHb) | 6.1 (5.5–6.7) | 5.8 (4.9–6.7) | 6.4 (5.4–7.4) | 6.4 (5.4–7.4) |
| Standard deviation | 1.9 | 1.9 | 2.1 | 2.1 |
| Median (95% CI; IU/gHb) | 6.4 (5.0–6.7) | 5.6 (4.9–6.8) | 6.6 (4.8–7.4) | 6.6 (4.8–7.4) |
| Range (IU/gHb) | 2.5–10.7 | 2.5–9.4 | 2.7–10.7 | 2.7–10.7 |
| **Newborns** | | | | |
| Number of cases | 216 | 86 | 130 | 121 |
| Mean (95% CI; IU/gHb) | 7.3 (6.9–7.7) | 7.6 (7.1–8.0) | 7.1 (6.5–7.7) | 7.6 (7.0–8.1) |
| Standard deviation | 2.9 | 1.9 | 3.4 | 2.9 |
| Median (95% CI; IU/gHb) | 7.8 (7.5–8.1) | 7.5 (7.1–8.1) | 8.0 (7.8–8.2) | 8.1 (7.8–8.5) |
| Range | 0.1–13.3 | 1.7–10.8 | 0.1–13.3 | 0.9–13.3 |

gHb. In newborns with an average age of 4 (4 ± 4 day), complete blood count (CBC) and G-6-PD enzyme activity were investigated. The CBC results showed that the RBC count, hemoglobin (Hb) and hematocrit (Hct) had statistically significant differences between males and females ($p<0.05$) (S1 Table). As shown in Table 1, no significant differences in G-6-PD enzyme activity were observed between females (7.6 ± 1.9 IU/gHb) and males (7.1 ± 3.4 IU/gHb) ($p = 0.235$). The AMM of newborns was 8.1 IU/gHb, representing 100% G-6-PD activity in the study population.

## Assessment of factors influencing G-6-PD enzyme activity

The effect of certain factors on G-6-PD activity measurement was assessed in 40 adults by quantitative SGT. G-6-PD enzymatic activity collected by finger-prick was compared to venous blood keeping in EDTA vacutainer tube. The results demonstrated that the mean of G-6-PD activity from venous blood (6.1 ± 1.9 IU/gHb, 95%CI: 5.5 to 6.7) was slightly higher than the finger-prick (5.6 ± 1.6 IU/gHb, 95%CI: 5.1 to 6.1) with no statistically significant difference ($p = 0.257$) (Fig 1A). Activity of G-6-PD enzyme rose significantly with increasing hematocrit (Hct) concentration, accounting for 4.5 ± 0.3, 6.4 ± 0.2, 7.9 ± 0.3 and 8.7 ± 0.2 IU/gHb for 30%, 40%, 50% and 60%, respectively (Fig 1B). A Tukey post hoc test revealed that G-6-PD activity showed a statistically significant difference among groups ($p<0.05$) (S2 Table). The results implied that hematocrit concentration is a dependent factor influencing G-6-PD activity detection.

The stability of G-6-PD enzymatic activity was evaluated after storage for 3 days at room temperature (temperature range from 25˚C to 30˚C) and 4˚C (Fig 2). G-6-PD activity was measured immediately (0), 1, 2, 6, 12, 24, 48, and 72 hours after blood collection. For the room temperature condition, the mean of G-6-PD enzyme activity was decreased gradually over time and fell to 5.1 ± 0.2 IU/gHb (95%CI: 4.7 to 5.5) at 72 hours (a mean fractional fall of 1.1 IU/gHb) (Fig 2A). On the other hand, there was no statistically significant difference when the blood was kept at 4˚C for 72 hours with mean activity of 6.1 ± 0.2 IU/gHb (95%CI: 5.7 to 6.4) and a mean fractional fall of 0.2 IU/gHb) (Fig 2B). A drop in G-6-PD activity was statistically correlated with its activity at baseline within 72 hours at room temperature and 4˚C ($p = 0.001$ and $p = 1.000$) (S3 Table). The regression analysis indicated that G-6-PD activity would decrease to 5.5 IU/gHb (10% from median 6.1 IU/gHb) for 2 days at room temperature and 182 days at 4˚C (Fig 2C and 2D, respectively). The results suggest that G-6-PD enzyme activity can remain stable at 4˚C for 3 days.

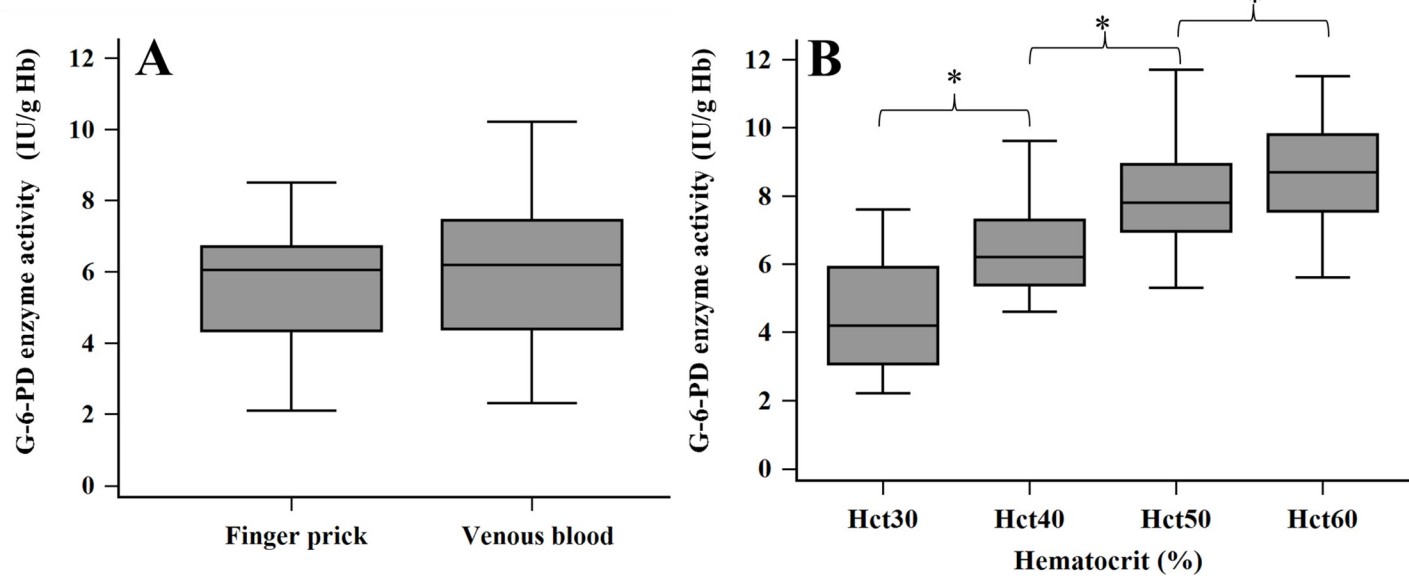

**Fig 1. Effect of G-6-PD enzyme activity detection on type of blood collection (A) and hematocrit concentration (B).** * indicates a significant difference among groups.

## Performance of the careSTART™ G6PD biosensor for G-6-PD enzyme activity detection

G-6-PD status was assessed in all 216 neonates by CBC, FST, careSTART™ G6PD biosensor and quantitative SGT. The mean delay between blood collection and laboratory processing was 22.4 hours (range from 12 to 26 hours). Efficacy evaluation involving the correlation analysis, sensitivity, specificity, positive likelihood ratio (+LR), negative likelihood ratio (-LR), positive predictive value (PPV) and negative predictive value (NPV), the hemoglobin (Hb) data from CBC results were applied to normalize the G-6-PD enzyme activity. Correlation analysis for G-6-PD enzyme activity measurement between the careSTART™ G6PD biosensor and SGT assay was very strong with Pearson's correlation coefficient of 0.82 (95%CI: 0.78 to 0.86, $p$ = 0.000) (Fig 3A). Intraclass correlation analysis between the two assays showed good agreement with the coefficient of 0.90 (95%CI: 0.87 to 0.92). The mean difference between the SGT and careSTART™ G6PD biosensor was 2.9 U/g Hb with 95% limit of agreement ranging from 6.6 to -0.8 U/g Hb (Fig 3B). The results indicated that 95% of the differences between the two assays were within this range.

The classification of G-6-PD deficiency relies on the guide of G-6-PD deficiency rapid diagnostic testing to support the curative treatment of malaria [14, 22] and WHO prequalification [24]. The receiver operator curve (ROC) analysis between the careSTART™ G6PD biosensor and SGT was considered at 30%, 70% and 80% G-6-PD activity (Fig 4). Analysis of the area under the ROC curve (AUC) was 0.982 (95%CI: 0.954 to 0.995), 0.993 (95%CI: 0.971 to 1.000) and 0.978 (95%CI: 0.948 to 0.993) in 30%, 70% and 80% activity, respectively. The comparison analysis of AUC showed a statistical difference between areas ($p < 0.001$) at all thresholds (Fig 4). The performance of the careSTART™ G6PD biosensor included sensitivity, specificity, +LR, -LR, PPV, NPV and disease prevalence, as illustrated in Table 2. A suitable cut-off was observed at 30% residual activity ($\leq$ 2.4 IU/gHb) with 100% sensitivity, 96% specificity, 28 of +LR, 73% PPV, 100% NPV, 97% accuracy and 8.8% disease prevalence.

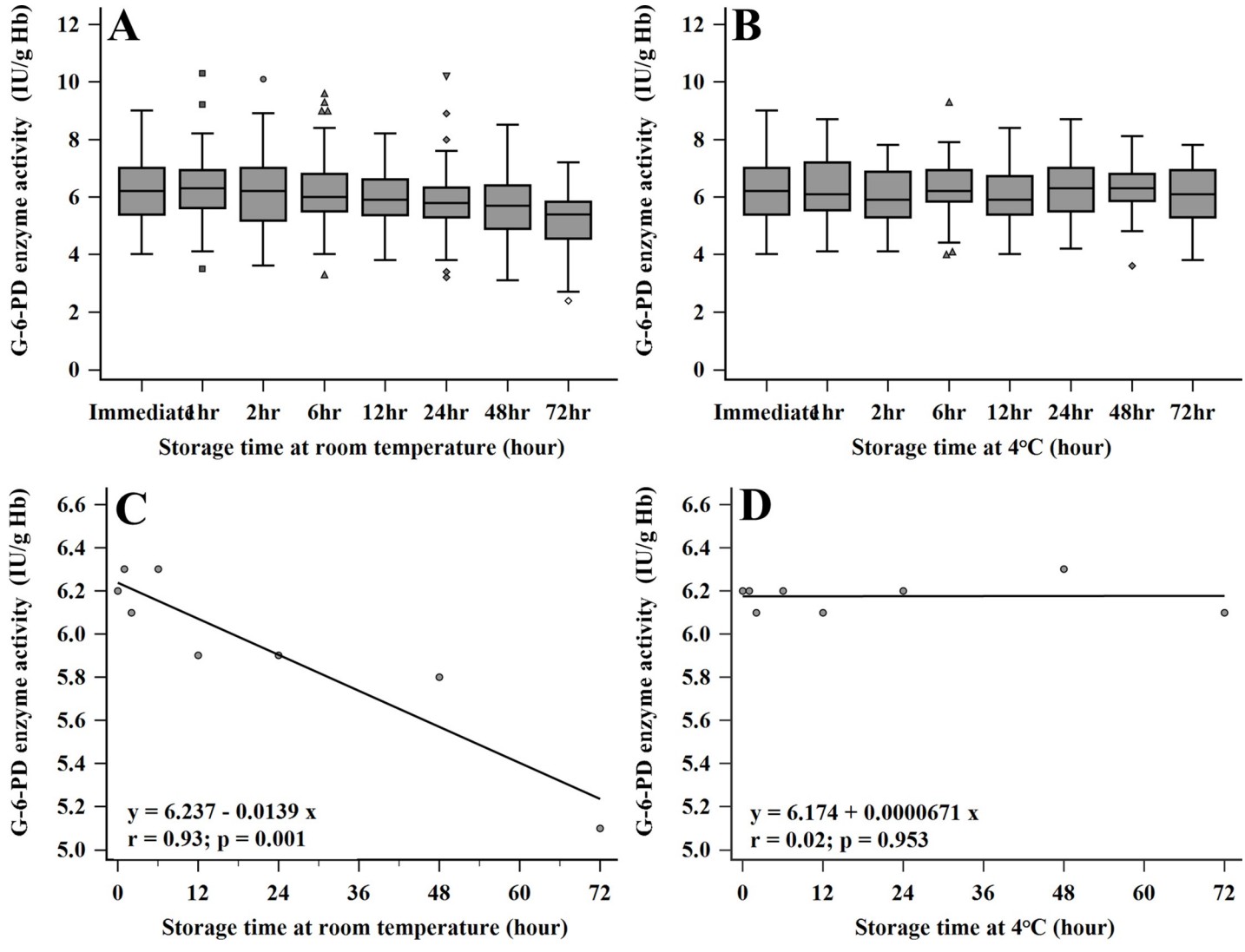

**Fig 2. Variation of G-6-PD activity detection after storage for 72 hours at room temperature (A) and 4˚C (B).** The ●, ■, ▲, and ♦ symbols indicate outlier data with less or greater than the interquartile range (<5ᵗʰ or >95ᵗʰ percentile). Regression analysis of G-6-PD activity detection after storage for 72 hours under room temperature (C) and 4˚C (D).

Assessment for the clinical diagnostic ability of the careSTART$^{TM}$ G6PD biosensor was analyzed with standard SGT by gender (Table 3). Normal G-6-PD enzyme activity was identified from AMM in the study population. The cut-off value was assigned at 30% and 80% according to the above mentioned. Individuals with G-6-PD enzyme activity less than 30% of AMM (< 2.4 IU/gHb) were identified as having G-6-PD deficiency. Considering female heterozygous having G-6-PD activity at 30% to 80% of AMM (2.4–6.5 IU/gHb) was defined as G-6-PD intermediate. Females with G-6-PD activity more than 80% of AMM (> 6.5 IU/gHb) were deemed normal. Sensitivity, specificity, accuracy, PPV and NPV analyzed by the careSTART$^{TM}$ G6PD biosensor were 89%, 93%, 92%, 67% and 98% in males, respectively, 64%, 83%, 80%, 43% and 92% in females, respectively, and 78%, 89%, 87%, 56% and 96% in total, respectively (Table 3). The results were considered with a commonly used qualitative FST. As shown in Table 4, the overall efficiency of FST was 18% sensitivity, 100% specificity, 88%

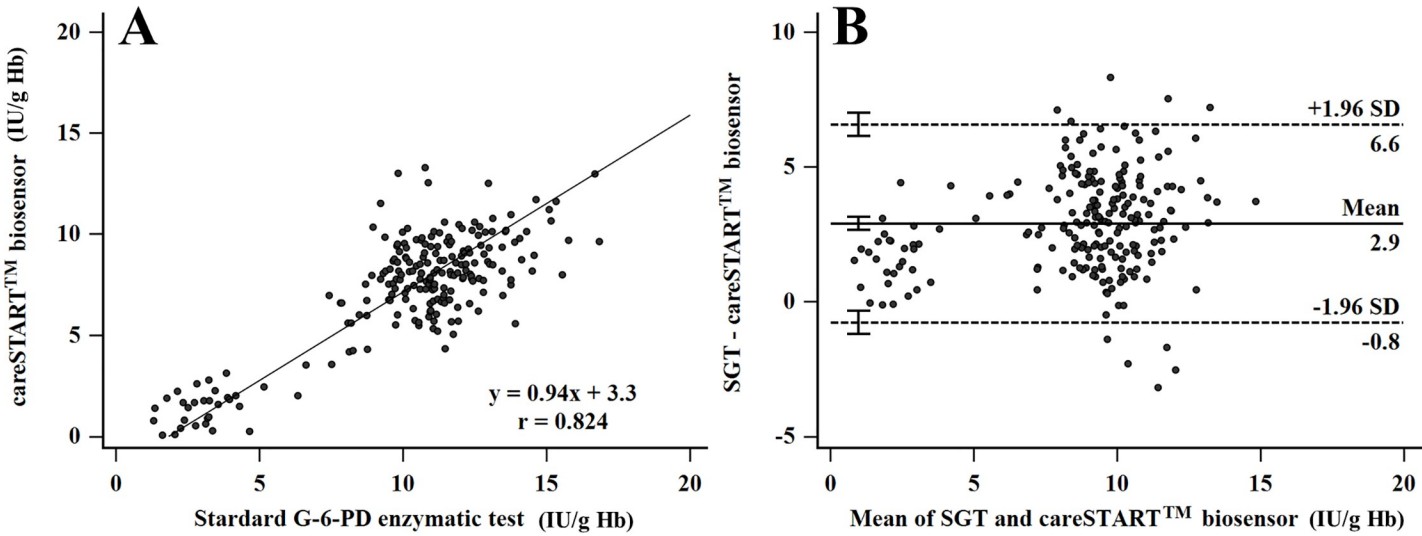

**Fig 3. Correlation analysis** between the careSTART[TM] G6PD biosensor and standard G-6-PD enzymatic test (SGT) analyzed by Pearson's correlation analysis (A) and Bland-Altman plot (B).

accuracy, 100% PPV and 88% NPV. These findings suggested that the careSTART[TM] G6PD biosensor possessed higher sensitivity than FST.

## Discussion

The detection of G-6-PD deficiency has been promoted to identify newborns in many countries [5, 6, 10, 25]. Early diagnosis is the most effective management strategy not only for correcting the problem of jaundice in newborns, but also making patients more aware of their deficiency. Boonpeng et al. reported that newborns in Thailand with more than 4.4 mg/dL of microbilirubin had a higher risk of G-6-PD deficiency [26]. This finding created a significant impact on the health policy and clinical management of newborns for improving the quality of

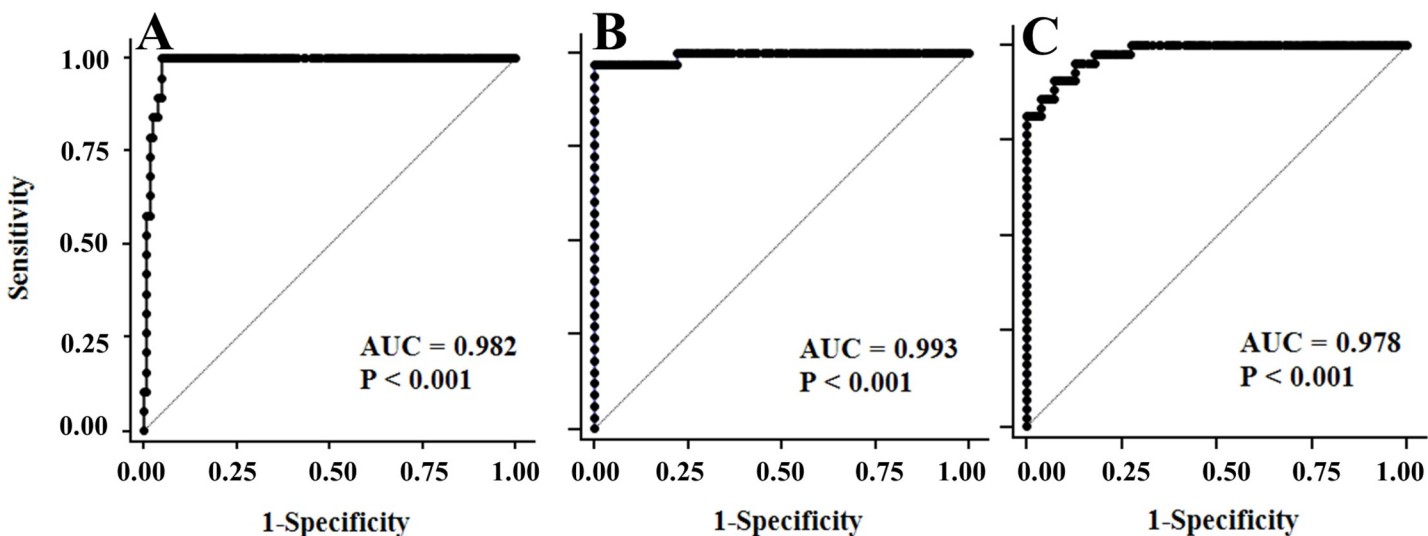

**Fig 4. Receiver Operator Characteristic (ROC) analysis of careSTART[TM] G6PD biosensor.** (biosensor) with standard G-6-PD enzymatic test (SGT) at 30% (A), 70% (B) and 80% (C) cut-off. AUC means the area under the ROC curve.

**Table 2. Performance of the careSTART[TM] G6PD biosensor for different residual G-6-PD activity.**

| Cut-off | G-6-PD activity cut-off value (IU/gHb) | Sensitivity (95%CI) | Specificity (95%CI) | Positive likelihood ratio (95%CI) | Negative likelihood ratio (95%CI) | Positive predictive value (95%CI) | Negative predictive value (95%CI) | Accuracy (95%CI) | Disease prevalence (95%CI) |
|---|---|---|---|---|---|---|---|---|---|
| 30% | <2.4 | 100% (82–100) | 96% (93–99) | 28 (14–58) | 0 | 73% (57–85) | 100% | 97% (93–99) | 8.8% (5–13) |
| 70% | ≤5.7 | 100% (89–100) | 93% (89–97) | 15 (9–26) | 0 | 73% (61–83) | 100% | 94% (91–97) | 15% (11–21) |
| 80% | ≤6.5 | 100% (92–100) | 91% (85–95) | 11 (7–17) | 0 | 73% (63–81) | 100% | 93% (88–96) | 20% (15–26) |

life and protecting the activities associated with life expectancy. Most G-6-PD deficient individuals should avoid the oxidative stress triggered by agents such as antimalarial drugs (primaquine, dapsone or tafenoquine), infection, or the ingestion of fava beans [1, 4, 7]. The FST assay is an easy, rapid and cost-effective qualitative method recommended by the International Committee for Standardization in Hematology (ICSH) for screening G-6-PD deficiency [5, 19]. However, the quantitative G-6-PD enzymatic activity assay (SGT) measured by spectrophotometric and automated UV enzymatic methods remains the reference method [5, 16, 25, 27]. The threshold of G-6-PD activity has also been used as criteria for malarial drug execution [7, 8, 11–13]. The WHO recommended that patients having less than 30% G-6-PD enzymatic activity and 70% normal activity should be excluded from primaquine and tafenoquine treatment [14, 22]. The SGT method requires a source of electricity, refrigeration for reagents and substrates, information on patient hemoglobin levels, a spectrophotometer to measure the

**Table 3. Diagnostic performance of the careSTART[TM] G6PD biosensor classified by gender.**

| careSTART[TM] G6PD biosensor | Standard quantitative G-6-PD test (SGT) | | | Total | Performance | % (95% CI) |
|---|---|---|---|---|---|---|
| | Deficiency (<3.2 IU/gHb) | Intermediate (3.3–8.8 IU/gHb) | Normal (>8.8 IU/gHb) | | | |
| **Male** | | | | | Sensitivity | 88.9 (65.3–98.6) |
| Deficiency[a] | 16 | *NA | 8 | 24 | Specificity | 92.9 (86.4–96.9) |
| Normal[c] | 2 | *NA | 104 | 106 | Accuracy | 92.3 (86.3–96.3) |
| Total | 18 | *NA | 112 | 130 | PPV | 66.7 (50.1–79.9) |
| Prevalence of G-6-PD deficiency (%) | | | | 13.9 | NPV | 98.1 (93.4–99.5) |
| **Female** | | | | | Sensitivity | 64.3 (35.1–87.2) |
| Deficiency[a] | 1 | 1 | 0 | 2 | Specificity | 83.3 (72.7–91.1) |
| Intermediate[b] | 0 | 7 | 12 | 19 | Accuracy | 80.2 (70.3–88.0) |
| Normal[c] | 0 | 5 | 60 | 65 | PPV | 42.9 (28.2–58.9) |
| Total | 1 | 13 | 72 | 86 | NPV | 92.3 (85.5–96.1) |
| Prevalence of G-6-PD deficiency (%) | | | | 16.3 | | |
| **Total** | | | | | Sensitivity | 78.1 (60.0–90.7) |
| Deficiency[a] | 17 | 1 | 8 | 26 | Specificity | 89.1 (83.7–93.2) |
| Intermediate[b] | 0 | 7 | 12 | 19 | Accuracy | 87.5 (82.3–91.6) |
| Normal[c] | 2 | 5 | 164 | 171 | PPV | 55.6 (44.3–66.3) |
| Total | 19 | 13 | 184 | 216 | NPV | 95.9 (92.4–97.8) |
| Prevalence of G-6-PD deficiency (%) | | | | 14.8 | | |

*Not available (NA)

The threshold of G6PD biosensor for identifying subjects as deficient ([a]) is less than < 2.4 IU/gHb, intermediate ([b]) is 2.4 to 6.5 IU/gHb and normal ([c]) is more than 6.5 IU/gHb.

**Table 4. Diagnostic performance of the FST classified by gender.**

| careSTART™ G6PD biosensor | Standard quantitative G-6-PD test (SGT) | | | | Performance | % (95% CI) |
|---|---|---|---|---|---|---|
| | Deficiency (<3.2 IU/gHb) | Intermediate (3.3–8.8 IU/gHb) | Normal (>8.8 IU/gHb) | Total | | |
| **Male** | | | | | Sensitivity | 27.8 (9.7–53) |
| Deficiency | 5 | *NA | 0 | 5 | Specificity | 100 (96.8–100.0) |
| Normal | 13 | *NA | 112 | 125 | Accuracy | 90.0 (83.5–94.6) |
| Total | 18 | *NA | 112 | 130 | PPV | 100 |
| Prevalence of G-6-PD deficiency (%) | | | | 13.9 | NPV | 89.6 (86.6–92.0) |
| **Female** | | | | | | |
| Deficiency | 1 | 0 | 0 | 1 | Sensitivity | 7.1 (0.2–33.9) |
| Intermediate | *NA | *NA | *NA | *NA | Specificity | 100 (95.0–100.0) |
| Normal | 0 | 13 | 72 | 85 | Accuracy | 84.9 (75.5–91.7) |
| Total | 1 | 13 | 72 | 86 | PPV | 100 |
| Prevalence of G-6-PD deficiency (%) | | | | 16.3 | NPV | 84.7 (82.7–86.5) |
| **Total** | | | | | | |
| Deficiency | 6 | 0 | 0 | 6 | Sensitivity | 18.8 (7.2–36.4) |
| Intermediate | *NA | *NA | *NA | *NA | Specificity | 100 (98.0–100.0) |
| Normal | 13 | 13 | 184 | 210 | Accuracy | 88.0 (82.9–92.0) |
| Total | 19 | 13 | 184 | 216 | PPV | 100 |
| Prevalence of G-6-PD deficiency (%) | | | | 14.8 | NPV | 87.6 (85.7–89.3) |

*Not available (NA)

absorbance and personal expertise to perform the test. These limitations restrict the SGT for routine testing and field application, especially in endemic countries with malaria infection [18]. The careSTART™ G6PD biosensor has been created due to the current trend of developing point-of-care diagnostics. It meets not only all of the point-of-care testing (POCT) requirements, but is also an interesting quantitative tool to identify G-6-PD deficiency in newborns. The careSTART™ G6PD biosensor used in the present research is a new model (version 2017), which has improved reliability for results by adding the reference electrode. The performance and factors influencing G-6-PD activity measurement were elucidated in this investigation. Basic information about G-6-PD activity in the study population was analyzed by gender (Table 1). There were no significant differences in G-6-PD activity between males and females. Moreover, the mean and AMM of G-6-PD enzyme activity in newborns were significantly higher than adults ($p < 0.05$), corresponding with previous study [27]. The factors affecting G-6-PD activity measurement were examined in 40 adults involving the type of blood collection, Hct concentration, storage time and temperature. The results found that the venous blood preserved in EDTA anticoagulant did not interfere with G-6-PD enzyme activity measurement compared to finger blood (Fig 1A). This finding confirmed the report by Roca-Feltrer et al. [28], which solved the question of von Fricken ME's studies [25]. Accordingly, the hematocrit in newborns has a higher concentration than adults [29, 30], corresponding to this study (42 ±3.7% in adults and 49±7.7% in newborns). This means that newborns should have higher G-6-PD activity than adults, which is consistent with the results in Fig 1B. The results suggest that the reference value of newborns should be differentiated from adults [31]. G-6-PD activity drops gradually as time progresses [6]. Because the acceptance decreasing rate of G-6-PD activity should be no more than 10% [6, 32], G-6-PD activity was reduced over acceptance value at room temperature for 50 hours (Fig 2A) and remained stable at 4°C for 3 days (Fig

2B). The regression analysis found that blood samples should be stored at 4˚C for 3 days (Fig 2D). Similar to a previous report, the G-6-PD activity did not fall by 10% until after 9 days of storage [6]. The performance evaluation of the careSTART$^{TM}$ G6PD biosensor was analyzed using 216 neonates. The results demonstrated a strong positive correlation between the careSTART$^{TM}$ G6PD biosensor and SGT (Fig 3). Thus, it can be described that both methods directly determine the kinetics of G-6-PD enzymatic activity reactions [18]. Bland-Altman plot analysis showed the absolute activity values of the same samples obtained from SGT, which were slightly higher than the careSTART$^{TM}$ G6PD biosensor (+2.9 IU/gHb). Contrary to the automated UV enzymatic assay, G-6-PD activity was significantly higher than that of the SGT method [16]. This results from the different detectors; the careSTART$^{TM}$ G6PD biosensor measures the electrons of $Fe^{3+}$ from the change of $NADP^+$ to NADPH reaction upon G-6-PD enzyme, meaning the electrons may be lost during operation. In this study, we set the cut-off point for G-6-PD deficiency at 30%, 70% and 80% of AMM based on previous suggestions [14, 20, 22]. The current thresholds used as exclusion criteria for primaquine treatment (<30% activity) and tafenoquine (<70% activity) [5, 28]. The diagnostic ability of the developed careSTART$^{TM}$ G6PD biosensor represented an ideal method compared to a reference SGT assay at all thresholds, especially at 30% cut-off (Fig 4 and Table 2). With ROC analysis, the results suggested the optimal cut-off value to provide appropriate diagnostic performance (Table 2). The indication is that individuals with G-6-PD enzyme activity less than 2.4 IU/gHb can be identified as having G-6-PD deficiency, while those with G-6-PD enzyme activity between 2.5 to 6.4 IU/gHb can be identified as having intermediate G-6-PD deficiency. As presented in this study, the main advantage for clinical use in quantitative point-of-care diagnostics utilizing the careSTART$^{TM}$ G6PD biosensor is to provide a more accessible way of defining G-6-PD activity at bedside and identifying individuals with intermediate G-6-PD deficiency, especially in terms of differentiating heterozygous females. The study also illustrates the sensitivity, specificity, PPV, NPV and accuracy of the careSTART$^{TM}$ G6PD biosensor and FST compared to standard SGT (Table 3 and Table 4). The diagnostic efficacy of the careSTART$^{TM}$ G6PD biosensor was found to be slightly higher than reported in the past [6]. Such variance in the results may be caused by the small-sized group and different population (216 neonates in the present study versus 900 adults in the previous study). Table 4 illustrates the low sensitive achievement by FST assay at all thresholds. This result confirmed that all current qualitative tests perform poorly [6, 17, 25, 28]. However, it has been recommended by ICSH as the appropriate qualitative method for screening G-6-PD deficiency in the field [5, 19]. The careSTART$^{TM}$ G6PD biosensor showed a high potential for identifying G-6-PD deficiency in newborns and partial G-6-PD deficiency in females, imposing a risk for primaquine the same as tafenoquine administration. Moreover, quantitative G-6-PD activity has more alternative, attractive methods than molecular analysis because some cases of intermediate or G-6-PD deficiency could not be identified in the mutation [1, 3, 33, 34]. More than 400 different mutations have been found in individuals with G-6-PD deficiency [1, 3, 33, 34]. Some cases contain the mutations in cis-acting regulatory sequences or in the non-coding region of the G-6-PD gene, which may interfere with its expression [16]. However, the results by the careSTART$^{TM}$ G6PD biosensor should be normalized with Hb concentration, especially in hemolytic anemia patients, because false positive results may occur.

## Conclusion

The careSTART$^{TM}$ G6PD biosensor has a high diagnostic ability to quantify G-6-PD enzymatic activity. It meets the needs for point-of-care testing and an appropriate setting for health promotion. Particularly, the careSTART$^{TM}$ G6PD biosensor is suitable for early identification

of intermediate and G-6-PD deficiency in newborns, which is the most effective strategy for the prevention of severe hemolytic anemia and the initiation of corrective treatment in a timely manner.

## Supporting information

**S1 Table. The CBC results of 216 neonates.** Results of RBC count, Hb and Hct in males were significantly higher than females.
(DOCX)

**S2 Table. Multiple comparisons of G-6-PD activity among hematocrit (Hct) concentration in 40 adults.** The G-6-PD enzyme activity at certain Hct was subtracted with another getting the different of G-6-PD activity. The mean difference of G-6-PD activity measures the absolute difference between the mean values in two groups.
(DOCX)

**S3 Table. Multiple comparisons of G-6-PD enzyme storage stability at room temperature and 4˚C for 72 hours.** The mean difference measures the absolute difference between the mean value in two groups.
(DOCX)

## Acknowledgments

We are grateful to thank the Thammasat University Hospital for supplying neonates blood samples.

## Author Contributions

**Conceptualization:** Khaimuk Changsri, Sirinart Chomean.

**Data curation:** Pairat Pengboon, Areenuch Thamwarokun, Sirinart Chomean.

**Formal analysis:** Pairat Pengboon, Chollanot Kaset, Sirinart Chomean.

**Funding acquisition:** Khaimuk Changsri, Sirinart Chomean.

**Investigation:** Pairat Pengboon.

**Methodology:** Khaimuk Changsri.

**Validation:** Sirinart Chomean.

**Writing – original draft:** Sirinart Chomean.

**Writing – review & editing:** Chollanot Kaset, Sirinart Chomean.

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
