## [Decision Letter · Decision Letter 0]

2 Sep 2019

PONE-D-19-22851

Evaluation of quantitative biosensor for glucose-6-phosphate dehydrogenase activity detection

PLOS ONE

Dear Dr Chomean,

Thank you for submitting your manuscript to PLoS ONE. After careful consideration, we felt that your manuscript requires revision, following which it can possibly be reconsidered. As quoted by the reviewers, a number of methodological concerns should be clarified otherwise it my compromise the manuscript. The results needs to be revised, and the MS should be revised by a native English speaker or professional language editing service. For your guidance, a copy of the reviewers' comments was included below.   

We would appreciate receiving your revised manuscript by September 30. To enhance the reproducibility of your results, we recommend that if applicable you deposit your laboratory protocols in protocols.io, where a protocol can be assigned its own identifier (DOI) such that it can be cited independently in the future. For instructions see: http://journals.plos.org/plosone/s/submission-guidelines#loc-laboratory-protocols

We look forward to receiving your revised manuscript.

Kind regards,

Luzia Helena Carvalho, Ph.D.

Academic Editor

PLOS ONE

Journal Requirements:

2. Please provide additional details regarding participant consent. In the ethics statement in the Methods and online submission information, please ensure that you have specified (1) whether consent was informed and (2) what type you obtained (for instance, written or verbal). We noted your study included minors, please state whether you obtained consent from parents or guardians. If the need for consent was waived by the ethics committee, please include this information.

3. Please remove your figures from within your manuscript file, leaving only the individual TIFF/EPS image files, uploaded separately.  These will be automatically included in the reviewers’ PDF.

Additional Editor Comments (if provided):

Reviewers' comments:

Reviewer's Responses to Questions

**Comments to the Author**

1. Is the manuscript technically sound, and do the data support the conclusions?

Reviewer #1: Partly

Reviewer #2: Partly

2. Has the statistical analysis been performed appropriately and rigorously? 

Reviewer #1: No

Reviewer #2: Yes

3. Have the authors made all data underlying the findings in their manuscript fully available?

Reviewer #1: No

Reviewer #2: No

4. Is the manuscript presented in an intelligible fashion and written in standard English?

Reviewer #1: No

Reviewer #2: No

5. Review Comments to the Author

Reviewer #1: The article from Pengboon and colleagues presents a very important topic with implications on the health and clinical management of newborns in Thailand and elsewhere. The manuscript in its current form presents a number of issues that needs to be addressed.

1. English language needs an accurate revision throughout the manuscript.

2. The abstract does not seem to be coherent with the results and should be revised.

3. Methods: in the abstract and introduction only newborns are mentioned but then in methods the authors say they have analyzed adult samples and newborn samples. The rationale is not very clear in this section and I wonder whether this laboratory part needs to be included in the manuscript (see more comments below).

More details need to be provided about the Biosensor used (since CareStart has produced different models over the years and even made software updates within the same Biosensor model). Probably the dates of when the study was carried out would be helpful. Calculation of G6PD activity in Iu/gHb should be used always and authors need to specify here how they calculated for the Biosensor.

More details need to be provided about the G6PD reference test: how was it performed, using which reagents, how many replicates per sample, how many and how frequently control samples were used, which temperature was used, etc.

4. The WHO classification cannot be used to classify individual enzymatic activities. The classification was developed in the 80’s to categorize G6PD mutations based on the residual enzymatic activities found in hemizygous males. In order to classify subjects as G6PD deficient (<30% normal activity), normal (>80% normal activity) or intermediate (30-80% normal activity), an adjusted male population median needs to be calculated according to Domingo et al 2013. The authors need to revise and delete the use of WHO classification throughout the manuscript.

5. Results: for the analysis of factors influencing G6PD activity, samples should have been analyzed by the gold standard reference test while they were only analyzed with the Biosensor under validation making the results difficult to assess and possibly unreliable. Also, I don’t think 6-months storage stability ca be extrapolated from a 72hour experiment so this part should be cut.

The BA plot shows that the 95%CI of the difference between the 2 assay is +6.6 and -0.8 with the mean difference in the 2 assay is 2.9U/gHb, a rather large difference for a test that has a range of 0-15U; in fact the 95%CI span for 7.4U, corresponding to almost 50% of the entire activity range. I am therefore surprised by the results of the AUC for ROC analysis as it seems to show much better performances of Biosensor than what showed by the BA plot. The tables and text show results for CareStart in terms of U/dL but the whole manuscript should present the quantitative G6PD data in a comparable way, ie.e IU/gHb. The authors need to revise Table 1 and 2 and especially explain Table 2 because it is not clear what the table is showing.

The direct comparison of FST with Biosensor is not appropriate because the FST is a qualitative test, Nonetheless, it makes sense to analyze the performances of both tests, but I would keep the results separate.

Reviewer #2: There are no page numbers or page lines making the review of this article difficult.

The language needs significant editing.

Will just refer to sections and sentences:

1. Results: Assessment of factors affecting for G-6-PD enzyme activity analysis. In the paragraph starting “Due to the newborn was classified…” this whole paragraph requires language editing. Presumably the first sentence refers to the fact that variable were study in adults since more blood could be collected from them?

2. “The results suggested that the G-6-PD enzyme activity can be stable at 4°C for 5 months.” Should be removed. This claim should be based on realtime data. There is no precedent for this level of stability.

3. Results: Performance of the careSTARTTM G6PD biosensor for G-6-PD enzyme activity detection “For efficacy evaluation…”.. The authors should explain what this means.

4. In the same section “ According to the adjusted male median, the receiver operator curve (ROC)…” the authors should clarify whether U/dL were used or U/g Hb values were used.

5. Figure 4. The plots are Carestart in U/g Hb, so these have been normalized.

6. Discussion: ..” WHO recommended that the patients that have G-6-PD enzymatic activity less than 30% and 70% of normal activity should be excluded from primaquine and tafenoquine treatment, respectively [5, 14, 16].” This statement is simply false, and the references have nothing to do with this statement.

7. Discussion: …”It proved the doubt messages of von Fricken ME’s studies [22] and consistent

with the Roca-Feltrer A. and coworker performing by CareStart G6PD RDT [14]” the authors should clarify what this means.

8. Discussion: ..” that blood samples should be stored at 4°C for 6 months.” Again this claim should not be made only that it is stable for 72 hours.

9. Discussion:..” The current thresholds used as exclusion criteria for primaquine treatment (<30% activity) and tafenoquine (<70% activity) [5, 14]” This statement is correct but references are incorrect: the authors should refer to WHO malaria guidelines and Llanos et al article on tafenoquine.

10. Discussion: The authors should highlight that a limitation of the product is that, as their data suggests the Carestart product does not correct for hematocrit, and this may be a source of error in a larger study population.

11. Discussion: “…showed a high diagnostic potency..” a high potential?

6. PLOS authors have the option to publish the peer review history of their article (what does this mean?). If published, this will include your full peer review and any attached files.

Reviewer #1: No

Reviewer #2: No

---

## [Author Response · Author response to Decision Letter 0]

4 Oct 2019

Response to Referees’ comments and detail changed.

Reviewer #1

1.

Referees’ comments

The article from Pengboon and colleagues presents a very important topic with implications on the health and clinical management of newborns in Thailand and elsewhere. The manuscript in its current form presents a number of issues that needs to be addressed.

Revised statements

I would appreciate proper recommending of clinical management of newborns in Thailand. The mentioned paper has been quoted in the discussion part. “Boonpeng P. and colleague reported that newborns with microbilirubin more than 4.4 mg/dL having a higher risk of G-6-PD deficiency in Thailand [26]. This finding has valuable to define the policy on health and clinical management of newborns for improving the quality of life and protecting the activity affect life expectancy.”

Location

Manuscript in Discussion part (page 15 line 5 to 8 and reference # 26)

2. 

Referees’ comments

English language needs an accurate revision throughout the manuscript.

Revised statements

We sincerely apologize for the grammar and typos. We have revised the manuscript.

Location

Throughout Manuscript

3. 

Referees’ comments

The abstract does not seem to be coherent with the results and should be revised.

Revised statements

We have revised the abstract and mentioned below.

Location

Manuscript: Abstract part

4.

Referees’ comments

Methods: in the abstract and introduction only newborns are mentioned but then in methods the authors say they have analyzed adult samples and newborn samples. The rationale is not very clear in this section and I wonder whether this laboratory part needs to be included in the manuscript (see more comments below).

More details need to be provided about the Biosensor used (since CareStart has produced different models over the years and even made software updates within the same Biosensor model). Probably the dates of when the study was carried out would be helpful. Calculation of G6PD activity in Iu/gHb should be used always and authors need to specify here how they calculated for the Biosensor.

More details need to be provided about the G6PD reference test: how was it performed, using which reagents, how many replicates per sample, how many and how frequently control samples were used, which temperature was used, etc.

Revised statements

Methods: in the abstract and introduction only newborns are mentioned but then in methods the authors say they have analyzed adult samples and newborn samples.

Author’s responses

The present study aimed to evaluate the G-6-PD activity in newborns. However, the blood samples collected from newborns should be as little as possible. Therefore, the factors affecting the G-6-PD activity was determined in adult blood.

Anyway, the abstract was edited and also added the methods as well as results of adults and newborns.

Location

Manuscript in Abstract part (page 2 line 7 to 10)

Manuscript in Materials and Methods (page 4 line 8 to 10)

Revised statements

More details need to be provided about the Biosensor used (since CareStart has produced different models over the years and even made software updates within the same Biosensor model).

Author’s responses

The careSTARTTM G6PD biosensor used in this study is a new model (2017) which modified the reference electrode to improve the reliability of the test. This detail was appended in discussion part. As the text “A careSTARTTM G6PD biosensor used in the present study is a new model (version 2017) which improved the reliability of results by adding the reference electrode.”

Location

Manuscript in Discussion part (page 15 line 24 to 26)

Revised statements

Calculation of G6PD activity in Iu/gHb should be used always and authors need to specify here how they calculated for the Biosensor.

Author’s responses

We accepted the reviewer’s comment. The unit of G6PD activity was changed in IU/gHb throughout the manuscript. The activity derived from biosensor measured the G-6-PD enzyme from RBC in one deciliter which should be divided by Hb (IU/gHb).

Location

Throughout Manuscript

Manuscript in Materials and Methods (page 5 line 29 to 30)

Fig. 1, 2 and 3 were changed.

Revised statements

More details need to be provided about the G6PD reference test: how was it performed, using which reagents, how many replicates per sample, how many and how frequently control samples were used, which temperature was used, etc.

Author’s responses

The method of a reference assay (SGT) and FST were explained in materials and methods.

For SGT assay, All experiments were performed by using the SGT assay according to the manufacturer’s protocol. Briefly, five microliters of blood was mixed with the Elution Buffer (75 µL) in U-bottom microtiter plate for 10 min at room temperature. The eluted (15 µL) was transfer to 75 µL of the Reagent Mixture containing in a new flat-bottom microtiter plate and mix thoroughly. Then, adding 100 µL of Color Reagent Mixture to each well and read the G-6-PD activity reaction (kinetic mode) with ELISA reader (800TS Microplate Reader, BioTek, USA) at 550 nm for 15 min with 1 min intervals. After the final reading is taken, the plate was read at wavelength 405 nm to get the hemoglobin (Hb) content of each sample. The G-6-PD enzyme activity is calculated by comparing the rates of blood sample to rate of normal control with known G-6-PD activity. The reported G-6-PD activity has been normalized to Hb and expressed results directly into international units per gram of hemoglobin (IU/gHb). Quality control of deficient blood (2.0 IU/g Hb with specification of 1.0-3.0 IU/gHb) and normal blood (14.6 IU/gHb with specification of 9.5-19.7IU/gHb) was operated throughout the experiments.”

For FST assay, “Five microliter of EDTA blood was added to 100 µL of G-6-PD substrate reagent, mixed thoroughly and incubated at 25°C for 30 min. After that the mixture (100 µL) was applied to filter paper and observed under UV light. Results accomplished by careSTARTTM G6PD biosensor and FST were compared with standard quantitative G-6-PD assay (SGT) as a reference assay.”

Location

Manuscript in Materials and Methods (page 5 line 10 to 22 and page 6 line 2 to 6) 

5.

Referees’ comments

The WHO classification cannot be used to classify individual enzymatic activities. The classification was developed in the 80’s to categorize G6PD mutations based on the residual enzymatic activities found in hemizygous males. In order to classify subjects as G6PD deficient (<30% normal activity), normal (>80% normal activity) or intermediate (30-80% normal activity), an adjusted male population median needs to be calculated according to Domingo et al 2013. The authors need to revise and delete the use of WHO classification throughout the manuscript.

Revised statements

We accepted the reviewer’s comments. So, the results were re-analyzed using the cut-off at 30% and 80% of residual G-6-PD enzyme activity. Moreover, the threshold at 70% was performed according to the suggestion of tafenoquine treatment. In addition, the adjusted male population median was calculated according to Domingo et al . (2013).

Location

Manuscript: Materials and Methods (statistical analysis) part. (page 6 line 11 to 16 and reference # 20 #21 #22)

6. 

Referees’ comments

Results: for the analysis of factors influencing G6PD activity, samples should have been analyzed by the gold standard reference test while they were only analyzed with the Biosensor under validation making the results difficult to assess and possibly unreliable. Also, I don’t think 6-months storage stability ca be extrapolated from a 72hour experiment so this part should be cut.

Revised statements

We accepted the reviewer’s comments and demonstrated the results from a standard SGT as a reference method in this study. Moreover, the stability of G-6-PD activity was concluded at 4°C for 3 days. 

Location

Manuscript: Materials and Methods part (page 4 line 24, page 5 line 10 to 22), Results part (page 9 line 12 to 13) and Discussion part (page 16 line 13 to 14)

7.

Referees’ comments

The BA plot shows that the 95%CI of the difference between the 2 assay is +6.6 and -0.8 with the mean difference in the 2 assay is 2.9U/gHb, a rather large difference for a test that has a range of 0-15U; in fact the 95%CI span for 7.4U, corresponding to almost 50% of the entire activity range. I am therefore surprised by the results of the AUC for ROC analysis as it seems to show much better performances of Biosensor than what showed by the BA plot. The tables and text show results for CareStart in terms of U/dL but the whole manuscript should present the quantitative G6PD data in a comparable way, ie.e IU/gHb. The authors need to revise Table 1 and 2 and especially explain Table 2 because it is not clear what the table is showing.

Revised statements

In order to evaluate whether the careSTARTTM G6PD biosensor and standard G-6-PD enzymatic test (SGT) can be used for classifying deficient, intermediate, and normal activity between the two assays, the Bland–Altman plot was used (Fig. 3). The measurements made by the SGT exceeded those obtained by careSTARTTM G6PD biosensor by an average of 2.9 IU/g Hb, indicates that value from careSTARTTM G6PD biosensor tends to be lower than those obtained by SGT. However, a significant correlation between the points on the Bland-Altman plot (Pearson’s correlation analysis, r=0.824; P<0.05), which provides evidence of a linear association. In conclusion, the Bland Altman plot provides only an agreement between the two methods.

To clarify “The BA plot shows that the 95%CI of the difference between the 2 assay is +6.6 and -0.8 with the mean difference in the 2 assay is 2.9U/gHb, a rather large difference for a test that has a range of 0-15U; in fact the 95%CI span for 7.4U, corresponding to almost 50% of the entire activity range” from reviewer’s comment. We would like to demonstrate some data from previous studies, Anantasomboon P. and coworkers [16] shows the 95%CI of the difference between the 2 assay is +3.21 and -7.28 with the mean difference in the 2 assay is -2.04U/gHb (spectrophotometric vs Mindray G6PD assay), the 95%CI span for 10.49 U/gHb. Conversely, the study from field evaluation [6] showed the mean differences: 0.83 U/gHb, 95%CI:-3.10-1.44 U/gHb (span for 4.54 U/gHb)

Due to the variation of the test, the 95%CI of the difference between the 2 assays may affect the sensitivity and specificity of the test. However, ROC analysis of careSTARTTM G6PD biosensor (biosensor) and fluorescent spot test (FST) with standard G-6-PD enzymatic test (SGT) indicate that AUC of careSTARTTM G6PD biosensor is greater than those fluorescent spot test (FST) Finally, and the optimal cut- off value can be determined using ROC curve analysis for careSTARTTM G6PD biosensor.

However, the ROC results were revised which analyzed between the biosensor and standard SGT (Fig. 4). Based on AUC analysis, the results also represented the optimal value for identifying the G-6-PD status at a certain cut-off (Table 2). 

Location

Manuscript: Results part (page 10 line 7 to 19; Fig 4 and Table 2)

8.

Referees’ comments

The direct comparison of FST with Biosensor is not appropriate because the FST is a qualitative test, Nonetheless, it makes sense to analyze the performances of both tests, but I would keep the results separate.

Revised statements

We agreed with reviewer’s comments and the results were separately illustrated. 

Location

Manuscript: Results part (page 12 line 1 to 13; Table 3 and Table 4)

Referee #2

1.

Referees’ comments

Results: Assessment of factors affecting for G-6-PD enzyme activity analysis. In the paragraph starting “Due to the newborn was classified…” this whole paragraph requires language editing. Presumably the first sentence refers to the fact that variable were study in adults since more blood could be collected from them?

Revised statements

We sincerely apologize for the grammar and typos.

The sentences were edited to “By the reason of blood drawing from a newborn can be difficult, potentially harmful and small volume, the factors affected to the G-6-PD enzyme activity measurement were evaluated in 40 adults” in Materials and Methods.

Location

Manuscript: Materials and Methods part (page 4 line 8 to 10)

2. 

Referees’ comments

“The results suggested that the G-6-PD enzyme activity can be stable at 4°C for 5 months.” Should be removed. This claim should be based on realtime data. There is no precedent for this level of stability.

Revised statements

We accepted the reviewer’s comments and demonstrated the stability of G-6-PD activity for 3 days. As the sentence “The results suggested that the G-6-PD enzyme activity can be stable at 4°C for 3 days.”

Location

Manuscript: Results part (page 9 line 12 to 13) and Discussion part (page 16 line 13 to 14)

3. 

Referees’ comments

Results: Performance of the careSTARTTM G6PD biosensor for G-6-PD enzyme activity detection “For efficacy evaluation…”.. The authors should explain what this means.

Revised statements

The efficacy evaluation involves assessing the performance of careSTARTTM G6PD biosensor comparable with standard SGT method. It included the correlation analysis, sensitivity, specificity, positive likehood ratio (+LR), negative likehood ratio (-LR), positive predictive value (PPV), negative predictive value (NPV) and disease prevalence.

Location

Manuscript: Results part (page 9 line 22 to 25)

4. 

Referees’ comments

In the same section “ According to the adjusted male median, the receiver operator curve (ROC)…” the authors should clarify whether U/dL were used or U/g Hb values were used.

Revised statements

We re-analyzed and the results were demonstrated only the unit of IU/gHb.

Location

Throughout Manuscript

5. 

Referees’ comments

Figure 4. The plots are Carestart in U/g Hb, so these have been normalized.

Revised statements

We re-analyzed and the results were demonstrated only the unit of IU/gHb.

Location

Figure 2.

6. 

Referees’ comments

Discussion: ..” WHO recommended that the patients that have G-6-PD enzymatic activity less than 30% and 70% of normal activity should be excluded from primaquine and tafenoquine treatment, respectively [5, 14, 16].” This statement is simply false, and the references have nothing to do with this statement.

Revised statements

We sincerely apologize and changed the references to [14] WHO. Testing for G6PD deficiency for safe use of primaquine in radical cure of P. vivax and P. ovale (Policy brief). Geneva: World Health Organization. 2016:28.and [22] WHO. Guide to G6PD deficiency rapid diagnostic testing to support P. vivax radical cure. Geneva: World Health Organization. 2018:34.

Location

Manuscript: Discussion part. (page 15 line 15 to 18 and reference #14 and #22)

7. 

Referees’ comments

Discussion: …”It proved the doubt messages of von Fricken ME’s studies [22] and consistent

with the Roca-Feltrer A. and coworker performing by CareStart G6PD RDT [14]” the authors should clarify what this means.

Revised statements

The results were correlated to the Roca-Feltrer A study[28] which can answer the question from von Fricken ME’s report [25]. This statement was rewritten to “This finding was consistent with the Roca-Feltrer A [(28)] and coworker report which solved the question of von Fricken ME’s studies (25).”

Location

Manuscript: Discussion part (page 16 line 4 to 5)

8. 

Referees’ comments

Discussion: ..” that blood samples should be stored at 4°C for 6 months.” Again this claim should not be made only that it is stable for 72 hours.

Revised statements

We agree with this statement and change the G-6-PD storage time for 3 days. As “The regression analysis found that that blood samples should be stored at 4°C for 3 days (Fig 2D).”

Location

Manuscript: Discussion part (page 16 line 13 to 14)

9. 

Referees’ comments

Discussion:..” The current thresholds used as exclusion criteria for primaquine treatment (<30% activity) and tafenoquine (<70% activity) [5, 14]” This statement is correct but references are incorrect: the authors should refer to WHO malaria guidelines and Llanos et al article on tafenoquine.

Revised statements

My apologies for the mistakes. The references were changed in [14] WHO. Testing for G6PD deficiency for safe use of primaquine in radical cure of P. vivax and P. ovale (Policy brief). Geneva: World Health Organization. 2016:28.and [22] WHO. Guide to G6PD deficiency rapid diagnostic testing to support P. vivax radical cure. Geneva: World Health Organization. 2018:34..

Location

Manuscript: References part [14] and [22]

10. 

Referees’ comments

Discussion: The authors should highlight that a limitation of the product is that, as their data suggests the Carestart product does not correct for hematocrit, and this may be a source of error in a larger study population.

Revised statements

The limitation of a point-of-care, careSTARTTM G6PD biosensor was added in the discussion as the sentence “However, the results by careSTARTTM G6PD biosensor should be normalized with Hb concentration especially in hemolytic anemia patients because the false positive result may occur.” 

Location

Manuscript: Discussion part (page 17 line 21 to 23)

11. 

Referees’ comments

Discussion: “…showed a high diagnostic potency..” a high potential?

Revised statements

We sincerely apologize for the grammar and typos. We have revised this sentence as “The careSTARTTM G6PD biosensor showed a high potential for identifying G-6-PD deficiency in newborns and partially G-6-PD deficiency in female and imposing a risk for primaquine as same as tafenoquine administration.”

Location

Manuscript: Discussion part (page 17 line 13 to 15)

Editor’s comments

1) Thank you for submitting your revised manuscript to PLOS ONE. Could you please make changes to the ethics statement that you have provided as follows:

- Please include the statement about consent in the Methods section of the manuscript (and not just on the online details form).

- In the Methods, please clarify whether the adults in the study provided written informed consent.

Revised statements

Prior to enrolment all volunteers provided written informed consent to participate and publication of results. This includes 20 males and 20 females with the age range of 18 to 24 years. After obtaining written informed consent, the blood was collected both finger-prick (5 µL) and venous puncture (3 mL in K2EDTA vacutainer tubes) and analyzed by standard quantitative G-6-PD enzymatic assay (R&D Diagnostics, Ltd., Greece).

Location

Manuscript: Materials and Methods part (page 4 line 10 to 15) 

- In the Methods, please clarify who waived informed consent from the parents/guardians of the neonates who provided blood samples, was informed consent waived by the IRB, or did the authors make this decision?

Revised statements

To evaluate the performance of careSTARTTM G6PD biosensor (WELLS BIO, INC., Korea), the 216 neonatal blood samples remaining from a routine laboratory testing were collected and kept in microtainer tube with K2EDTA anticoagulant. The neonatal blood samples used in this study were leftover samples following routine analysis and all data were de-identified before access. For these reasons, the ethics committee provided an waiver of informed consent.

Location

Manuscript: Materials and Methods part (page 4 line 16 to 20)

---

## [Decision Letter · Decision Letter 1]

23 Oct 2019

PONE-D-19-22851R1

Evaluation of quantitative biosensor for glucose-6-phosphate dehydrogenase activity detection

PLOS ONE

Dear Dr Chomean,

Thank you for resubmitting your manuscript to PLoS ONE. After careful consideration, we felt that your study has the potential to be published if it is revised to address fundamental point raised now by the reviewer. As quoted by the reviewers, the authors should clarify some specific topics related to methods and tables. At this time, we strongly suggest that the manuscript should be revised by a native English-speaker or a professional language editing service.

We would appreciate receiving your revised manuscript by  November 10. To enhance the reproducibility of your results, we recommend that if applicable you deposit your laboratory protocols in protocols.io, where a protocol can be assigned its own identifier (DOI) such that it can be cited independently in the future. For instructions see: http://journals.plos.org/plosone/s/submission-guidelines#loc-laboratory-protocols

We look forward to receiving your revised manuscript.

Kind regards,

Luzia Helena Carvalho, Ph.D.

Academic Editor

PLOS ONE

Reviewers' comments:

Reviewer's Responses to Questions

**Comments to the Author**

1. If the authors have adequately addressed your comments raised in a previous round of review and you feel that this manuscript is now acceptable for publication, you may indicate that here to bypass the “Comments to the Author” section, enter your conflict of interest statement in the “Confidential to Editor” section, and submit your "Accept" recommendation.

Reviewer #1: (No Response)

Reviewer #2: All comments have been addressed

2. Is the manuscript technically sound, and do the data support the conclusions?

Reviewer #1: Partly

Reviewer #2: Yes

3. Has the statistical analysis been performed appropriately and rigorously? 

Reviewer #1: Yes

Reviewer #2: Yes

4. Have the authors made all data underlying the findings in their manuscript fully available?

Reviewer #1: No

Reviewer #2: Yes

5. Is the manuscript presented in an intelligible fashion and written in standard English?

Reviewer #1: No

Reviewer #2: No

6. Review Comments to the Author

Reviewer #1: The manuscript has improved considerably from its first version. The authors have addressed most of my previous requests and comments. I still have some questions for the authors and they should clarify a number of issues.

The G6PD reference test: the authors report the protocol but they do not mention whether the test was performed at controlled temperature and whether samples were analyzed in duplicate. The authors should also report the manufacturer name.

The FST is not performed according to manufacturers’ instructions. Why the authors chose to incubate the blood and reagent mix for 30 minutes? I expect this to have a pretty big impact on the results of the test, in particular toward misclassification of deficient/intermediate samples as normal.

Table 2: I do not understand what this table is showing. The se and sp at the 3 different threshold is better using the values of the second column as compared to the “optimal cut-off value” of the 10th column. This needs to be explained in further details

The English language still needs a lot of work. There are many parts of the manuscript that are difficult to read; in some parts the actual meaning of what the authors wrote is not very clear. It is important that the authors improve the language throughout the manuscript to make it understandable.

Reviewer #2: The authors have been very diligent about responding to the reviewers comments and have actually presented the data in a clear manner.

The English in the manuscript still requires correcting. I have just focused on the abstract, but the rest of the manuscript probably needs further review:

1. line 12"...enzyme activity inspected that the activity determined from finger-prick.." replace "inspected" with "showed"

2. Line 13: " However G6PD activity has significantly increased in higher hematocrit concentration". maybe replace with "The G6PD activity value was highly dependent on the hematocrit, and increased with increasing hematocrit"

3. line 16: "...strongly correlated with Pearson’s " replace with "..a strong correlation with a Peasron's..."

4. line 16 replace "perfectly" with "perfect"

5.lines 17-19 define what G6PD range this was accurate for: G6Pd deficient?

The rest of the manuscript should also go further review for English language. Although the effort put by the authors is greatly appreciated.

7. PLOS authors have the option to publish the peer review history of their article (what does this mean?). If published, this will include your full peer review and any attached files.

Reviewer #1: No

Reviewer #2: No

---

## [Author Response · Author response to Decision Letter 1]

4 Nov 2019

(R2): Response to Referees’ comments and detail changed. 

Reviewer #1

1.

Referees’ comments

The G6PD reference test: the authors report the protocol but they do not mention whether the test was performed at controlled temperature and whether samples were analyzed in duplicate. The authors should also report the manufacturer name.

Author’s responses

Thank you for reviewer’s suggestion. I have added more information about your comments. All clinical blood samples were studied by whether FST, careSTARTTM G6PD biosensor or SGT assay at single time. Whereas, the control blood samples including deficient and normal blood was determined by FST, careSTARTTM G6PD biosensor and SGT assay in duplicate. Moreover, the manufacturer name was also added according to R&D Diagnostics, Ltd., Greece.

Revised statements

All blood samples were carried out standard operating procedures under controlled time and temperature followed the manufacturer's instruction. All experiments were conducted the quality control of deficient blood (2.0 IU/g Hb with the specification of 1.0-3.0 IU/gHb) and normal blood (14.6 IU/gHb with the specification of 9.5-19.7IU/gHb in duplicate.

Location

Manuscript in Materials and Methods part (page 4 line 26 to 27 and page 5 line 26 to 28)

2. 

Referees’ comments

The FST is not performed according to manufacturers’ instructions. Why the authors chose to incubate the blood and reagent mix for 30 minutes?

Author’s responses

I sincerely apologize for having this mistake. I confirmed the FST protocol that the EDTA blood and G-6-PD substrate reagent was incubated at 25°C for 10 minutes.

Revised statements

Five microliters of EDTA blood was added to 100 µL of G-6-PD substrate reagent, mixed thoroughly and incubated at 25°C for 10 min.

Location

Manuscript in Materials and Methods part (page 6 line 9)

3. 

Referees’ comments

Table 2: I do not understand what this table is showing. The se and sp at the 3 different thresholds is better using the values of the second column as compared to the “optimal cut-off value” of the 10th column. This needs to be explained in further details

Author’s responses

The purpose in Table 2 needs to represent the sensitivity, specificity, PPV, NPV and accuracy at different cut-off value. The cut-off value and optimal cut-off value of G-6-PD activity was calculated by a certain residual G-6-PD activity and statistical SPSS analysis, respectively. 

The selection of which cut-off value depends on the purpose of the G-6-PD detection. If the careSTARTTM G6PD biosensor users require to confirm G-6-PD deficiency, they should be chosen the cut-off value at 30% (<2.4 U/gHb) due to the performance of biosensor at this point gave a highest specificity (96%) compared to another. However, the performance of “optimal cut-off value” was lower than that of the “cut-off value” as reviewer’s comment; I deleted the content in “optimal cut-off value”.

Location

Manuscript: Results part and Table 2 (page 11)

4.

Referees’ comments

The English language still needs a lot of work. There are many parts of the manuscript that are difficult to read; in some parts the actual meaning of what the authors wrote is not very clear. It is important that the authors improve the language throughout the manuscript to make it understandable.

Author’s responses

I would like to apologize for making you uncomfortable. I take full responsibility to improve this manuscript. The English was proofread and edited by native English teacher.

Location

Throughout Manuscript

Referee #2

1.

Referees’ comments

The English in the manuscript still requires correcting.

Author’s responses

I'm very thankful for your suggestion. Also, I would like to apologize for making you uncomfortable. I take full responsibility to improve this manuscript. The English was proofread and edited by native English teacher.

Location

Throughout Manuscript

---

## [Decision Letter · Decision Letter 2]

19 Nov 2019

PONE-D-19-22851R2

Evaluation of quantitative biosensor for glucose-6-phosphate dehydrogenase activity detection

PLOS ONE

Dear Dr  Chomean,

Thank you for resubmitting your manuscript to PLoS ONE. After careful consideration, we felt that your manuscript still requires substantial revision, following which it can possibly be reconsidered. While the subject of the MS was of interest of the reviewers, relevant topics remain to be addressed.  More specifically, the authors should clarify about discrepant results between table 2 and 3 and include in the methods the information about single replicate to the reference G6PD test. As quoted by the reviewer, the manuscript language has only marginally improved At this time, we strongly recommend that the authors include the modifications requested by the reviewer, and the MS should be revised by a native English-speaker  or a professional language editing service.

We would appreciate receiving your revised manuscript by December 10. To enhance the reproducibility of your results, we recommend that if applicable you deposit your laboratory protocols in protocols.io, where a protocol can be assigned its own identifier (DOI) such that it can be cited independently in the future. For instructions see: http://journals.plos.org/plosone/s/submission-guidelines#loc-laboratory-protocols

We look forward to receiving your revised manuscript.

Kind regards,

Luzia Helena Carvalho, Ph.D.

Academic Editor

PLOS ONE

Reviewers' comments:

Reviewer's Responses to Questions

**Comments to the Author**

1. If the authors have adequately addressed your comments raised in a previous round of review and you feel that this manuscript is now acceptable for publication, you may indicate that here to bypass the “Comments to the Author” section, enter your conflict of interest statement in the “Confidential to Editor” section, and submit your "Accept" recommendation.

Reviewer #1: (No Response)

2. Is the manuscript technically sound, and do the data support the conclusions?

Reviewer #1: Partly

3. Has the statistical analysis been performed appropriately and rigorously? 

Reviewer #1: No

4. Have the authors made all data underlying the findings in their manuscript fully available?

Reviewer #1: No

5. Is the manuscript presented in an intelligible fashion and written in standard English?

Reviewer #1: No

6. Review Comments to the Author

Reviewer #1: 1. After the modifications in the latest version there are still some important issues to address:

- I do not see how the results on sensitivity, specificity, PPV and NPV presented in Table 2 (and abstract) and Table 3 are compatible. The 2 tables give different results and it is unclear which ones are correct.

- In the results it is mentioned that newborn age ranged from 1 to 35 days, this should really be clarified.

- The fact that the reference G6PD test was run only in single replicate needs to be made explicit in the methods, not only in the response to the reviewers questions.

2. The English language in the manuscript has only marginally improved. There are still many sentences that are difficult to understand.

7. PLOS authors have the option to publish the peer review history of their article (what does this mean?). If published, this will include your full peer review and any attached files.

Reviewer #1: No

---

## [Author Response · Author response to Decision Letter 2]

26 Nov 2019

(R3): Response to Referees’ comments and detail changed. 

Reviewer #1

1.

Referees’ comments

I do not see how the results on sensitivity, specificity, PPV and NPV presented in Table 2 (and abstract) and Table 3 are compatible. The 2 tables give different results and it is unclear which ones are correct.

Author’s responses

The results in table 2 and 3 presented in different purpose.

The results depicted in Table 2 were analyzed by the receiver operating characteristic (ROC) curve which performed by using the 30%, 70% and 80% of G-6-PD residual activity. The results obtained from the area under the Receiver Operating Curve (AUC) suggested the optimal cut-off value indicating the proper sensitivity, specificity, positive likelihood ratio (+LR), negative likelihood ratio (-LR), positive predictive value (PPV) and negative predictive value (NPV). Also, it was analyzed by G-6-PD residual activity with no consideration about gender.

 For table 3 and 4, the results illustrated the diagnostic potency of the careSTARTTM G6PD biosensor and FST by using 2x2 table. All data were indicated the G-6-PD deficiently status according to the description of Domingo GJ et al. [20]. So, we enter the number of cases in the diseased group that test positive (a) and negative (b); and the number of cases in the non-diseased group that test positive (c) and negative (d) [23].

For example, Sensitivity= a / (a+b)

Specificity= d / (c+d)

Positive likelihood ratio= True positive rate / False positive rate = Sensitivity / (1-Specificity)

Negative likelihood ratio= False negative rate / True negative rate = (1-Sensitivity) / Specificity

Revised statements

Abstract: Analysis of the area under the Receiver Operating Curve (AUC) illustrated that the careSTARTTM G6PD biosensor had 100% sensitivity, 96% specificity, 73% positive predictive value (PPV), 100% negative predictive value (NPV) and 97% accuracy at 30% of residual activity. While the diagnostic ability for identifying G-6-PD deficiency had 78% sensitivity, 89% specificity, 56% positive predictive value (PPV), 96% negative predictive value (NPV) and 88% accuracy when stratified by gender.

Table 3: Diagnostic performance of the careSTARTTM G6PD biosensor classified by gender

Table 4. Diagnostic performance of the FST classified by gender

Location

Manuscript in Abstract part (page 2 line 17 to 22) and results part (page 13 line 1 and page 14 line 1)

2. 

Referees’ comments

In the results it is mentioned that newborn age ranged from 1 to 35 days, this should really be clarified.

Author’s responses

We need to explain the descriptive data analysis in newborns.

Revised statements

Materials and Methods: To evaluate the performance of the careSTARTTM G6PD biosensor (WELLS BIO, INC., Korea), 216 neonatal blood samples ranging in age between 1 to 35 years old were collected and kept in a microtainer tube with K2EDTA anticoagulant.

Results: In newborns with an average age of 4 (4 ± 4 day), complete blood count (CBC) and G-6-PD enzyme activity were investigated.

Location

Manuscript in Materials and Methods parts (page 4 line 19) and Results part (page 7 line 20 and 22)

3. 

Referees’ comments

The fact that the reference G6PD test was run only in single replicate needs to be made explicit in the methods, not only in the response to the reviewers questions.

Author’s responses

We explained this comment in the Materials and Methods parts as following.

Revised statements

All blood samples were carried out a single time following standard operating procedures under controlled time and temperature.

Location

Manuscript in Materials and Methods part (page 4 line 27 and 28)

4. 

Referees’ comments

The English language in the manuscript has only marginally improved. There are still many sentences that are difficult to understand.

Author’s responses

The English language entire the manuscript was proofread and edited by native English speaker.

Location

Throughout the manuscript.

---

## [Decision Letter · Decision Letter 3]

4 Dec 2019

PONE-D-19-22851R3

Evaluation of quantitative biosensor for glucose-6-phosphate dehydrogenase activity detection

PLOS ONE

Dear Dr Chomean,

Thank you for submitting your manuscript for review to PLoS ONE. After careful consideration, we feel that your manuscript will likely be suitable for publication if it is revised to address a few point raised by the reviewer. Basically, the authors should adjust Table to include the threshold used for the Biosensor.

We would appreciate receiving your revised manuscript by December 10. To enhance the reproducibility of your results, we recommend that if applicable you deposit your laboratory protocols in protocols.io, where a protocol can be assigned its own identifier (DOI) such that it can be cited independently in the future. For instructions see: http://journals.plos.org/plosone/s/submission-guidelines#loc-laboratory-protocols

We look forward to receiving your revised manuscript.

Kind regards,

Luzia Helena Carvalho, Ph.D.

Academic Editor

PLOS ONE

Reviewers' comments:

Reviewer's Responses to Questions

**Comments to the Author**

1. If the authors have adequately addressed your comments raised in a previous round of review and you feel that this manuscript is now acceptable for publication, you may indicate that here to bypass the “Comments to the Author” section, enter your conflict of interest statement in the “Confidential to Editor” section, and submit your "Accept" recommendation.

Reviewer #1: All comments have been addressed

2. Is the manuscript technically sound, and do the data support the conclusions?

Reviewer #1: Yes

3. Has the statistical analysis been performed appropriately and rigorously? 

Reviewer #1: Yes

4. Have the authors made all data underlying the findings in their manuscript fully available?

Reviewer #1: No

5. Is the manuscript presented in an intelligible fashion and written in standard English?

Reviewer #1: Yes

6. Review Comments to the Author

Reviewer #1: I would like to thank the authors for addressing my requests. I wanted to specify further my question about Table 3; Since both the Biosensor and the reference test are quantitative tests (with continuous data results), in order to identify subjects as "deficient" or "normal" one needs to establish a threshold under which a sample is considered deficient and over which a sample is considered normal. So I would invite the authors to just specify somewhere in the Table or legend, what is the threshold they have used for the reference test and for the Biosensor.

7. PLOS authors have the option to publish the peer review history of their article (what does this mean?). If published, this will include your full peer review and any attached files.

Reviewer #1: No

---

## [Author Response · Author response to Decision Letter 3]

5 Dec 2019

(R4): Response to Referees’ comments and detail changed. 

Reviewer #1

1.

Referees’ comments

I wanted to specify further my question about Table 3; Since both the Biosensor and the reference test are quantitative tests (with continuous data results), in order to identify subjects as "deficient" or "normal" one needs to establish a threshold under which a sample is considered deficient and over which a sample is considered normal. So I would invite the authors to just specify somewhere in the Table or legend, what is the threshold they have used for the reference test and for the Biosensor.

Author’s responses

Thank you for reviewer’s comments. To clarify the data, we specify the threshold of G6PD biosensor with the legend (a, b and c). Whereas the reference value by SGT assay was mentioned in the Table 3 and 4.

Revised statements

For SGT assay, we addressed as “Deficiency” (<3.2 IU/gHb), “Intermediate” (3.3–8.8 IU/gHb) and “Normal” (>8.8 IU/gHb) in Table 3 and 4.

For G6PD biosensor, we addressed as “The threshold of G6PD biosensor for identifying subjects as deficient (a) is less than < 2.4 IU/gHb, intermediate (b) is 2.4 to 6.5 IU/gHb and normal (c) is more than 6.5 IU/gHb.”

Location

Manuscript: Results part Table (page 13-15).

---

## [Editor Report · Decision Letter 4]

10 Dec 2019

Evaluation of quantitative biosensor for glucose-6-phosphate dehydrogenase activity detection

PONE-D-19-22851R4

Dear Dr. Chomean,

We are pleased to inform you that your manuscript has been judged scientifically suitable for publication and will be formally accepted for publication once it complies with all outstanding technical requirements.

With kind regards,

Luzia Helena Carvalho, Ph.D.

Academic Editor

PLOS ONE
---

## [Editor Report · Acceptance letter]

12 Dec 2019

PONE-D-19-22851R4 

Evaluation of quantitative biosensor for glucose-6-phosphate dehydrogenase activity detection 

Dear Dr. Chomean:

I am pleased to inform you that your manuscript has been deemed suitable for publication in PLOS ONE. Congratulations! Your manuscript is now with our production department. 

With kind regards,

on behalf of

Dr. Luzia Helena Carvalho 

Academic Editor

PLOS ONE